# Effect of Hydroxyapatite Nanoparticles and Nitrogen Plasma Treatment on Osteoblast Biological Behaviors of 3D-Printed HDPE Scaffold for Bone Tissue Regeneration Applications

**DOI:** 10.3390/ma15030827

**Published:** 2022-01-21

**Authors:** Hyunchul Park, Jaeyoung Ryu, Seunggon Jung, Hongju Park, Heekyun Oh, Minsuk Kook

**Affiliations:** 1Bitgoeul Dental Clinic 2F, Doklibro 8 Nam-gu, Gwangju 61660, Korea; nestor99@naver.com; 2Department of Oral and Maxillofacial Surgery, School of Dentistry, Chonnam National University, Gwangju 61186, Korea; ryu@jnu.ac.kr (J.R.); seunggon.jung@jnu.ac.kr (S.J.); omspark@jnu.ac.kr (H.P.); hkoh@jnu.ac.kr (H.O.)

**Keywords:** high density polyethylene, 3D scaffold, 3D printing, plasma surface treatment, bone substitute

## Abstract

The need for the repair of bone defects has been increasing due to various causes of loss of skeletal tissue. High density polyethylenes (HDPE) have been used as bone substitutes due to their excellent biocompatibility and mechanical strength. In the present study, we investigated the preosteoblast cell proliferation and differentiation on the adding nano-hydroxyapatite (n-HAp) particles into HDPE scaffold and treating HDPE/n-HAp scaffolds with nitrogen (N_2_) plasma. The three-dimensional (3D) HDPE/n-HAp scaffolds were prepared by fused modeling deposition 3D printer. The HDPE/n-HAp was blended with 10 wt% of n-HAp particle. The scaffold surface was reactive ion etched with nitrogen plasma to improve the preosteoblast biological response in vitro. After N_2_ plasma treatment, surfaces characterizations were investigated using Fourier transform infrared spectroscopy, scanning electron microscopy, and atomic force microscopy. The proliferation and differentiation of preosteoblast (MC3T3-E1) cells were evaluated by MTT assay and alkaline phosphatase (ALP) activity. The incorporation of n-HAp particles and N_2_ plasma surface treatment showed the improvement of biological responses of MC3T3-E1 cells in the HDPE scaffolds.

## 1. Introduction

Recently, the demand for bone reconstruction has been increasing due to cancer surgery, congenital deformity, fractures, oral and maxillofacial surgery, and various causes of bone defect [1,2,3]. Autologous bone, which is most ideal as bone substitute materials, has disadvantages including morbidity of harvest site, short supply, and risk of significant resorption [4]. Hence, synthetic grafting materials have been extensively studied to overcome the obstacles of autografts, allografts [5,6]. Many synthetic materials such as bioactive glasses, glass ionomers, aluminum oxide, calcium sulfate, calcium phosphates, α- and β-tricalcium phosphate (TCP), and synthetic hydroxyapatite are currently used as bone grafting materials [7]. 

High density polyethylene (HDPE) has been widely used in medical implants because it is a porous synthetic polymer that is biologically inert and non-absorbable in the body. Due to these advantages, HDPE facial implants are widely used as bone substitutes in facial surgery [8,9,10]. For example, the porous HDPE (Medpor^®^, Stryker, Kalamazoo, MI, USA) provides the highly stable and somewhat flexible framework composed of interconnected pores and Medpor stimulates tissue to infiltrate into its pores [8,9,10]. 

Hydroxyapatite (Ca_10_(PO_4_)_6_(OH)_2_, HAp) is one of the bioceramics with inorganic components and structures close to bones and teeth, and therefore they have excellent biocompatibility with the bone tissue. Furthermore, HAp have osteoconductive properties and low resorption rate therefore they can be extensively used as a bone graft material [11,12,13]. However, HAp has the drawback of being brittle and fracture prone when a shock is loaded. Generally, the main drawback of bioceramics is their low fracture toughness and brittleness.

To overcome the disadvantage, synthetic polymers and bioceramics-based scaffolds were extensively studied [14,15,16]. Larrañaga and coworkers have reported that the addition of bioactive glass particles to poly (L-lactide) (PLLA) and poly (ε-caprolactone) (PCL) scaffolds sustains adipose-derived stem cells (ASCs) osteogenic differentiation, promote calcification, and induces the formation of a hydroxyapatite layer on the polymer scaffolds surface [15]. Haaparanta et al. demonstrated that porous surface and β-TCP particles on the polylactide/β-tricalcium phosphate (PLA/β-TCP) composite scaffolds may encourage the growth of bone cells [16].

In recent years, the 3D printing technique in the fabrication of polymer/bioceramic 3D scaffolds is attracting attention due to the advantages such as controllable pore size and porosity, high interconnectivity of pores, and free structural design [17,18,19].

Salmoria and coworkers have reported that HAp content of HDPE/HAp functional graded scaffold fabricated by selective laser sintering (SLS) could control microstructure and mechanical properties of scaffold [20]. Suwanprateeb et al. have reported that microstructure and tensile properties of heat-treated three-dimensional printing (3DP) HDPE bone implant were significantly influenced by the polyethylene content and heat treatment steps [21]. Kim et al. demonstrated that porous HDPE/poly (ethylene-co-acrylic acid) (PEAA) scaffold surface modified with collagen has enhanced the adhesion and proliferation of osteoblast cells [22]. As mentioned above, research on in vitro study of 3D HDPE/n-HAp composite scaffold fabricated by fused deposition modeling (FDM) 3D printing has not been reported yet.

Plasma surface modification of biomaterials could provide the new surface characteristics of polymeric implants to improve biocompatibility without changing the bulk properties [23]. Furthermore, this process also leads to changes in surface topologies and surface chemistry of the modified polymers [24].

Davoodi and coworkers demonstrated that contact angle was significantly decreased and surface roughness was increased after nitrogen plasma was treated on the PLA film. In addition, attachment and proliferation of L929 fibroblast cell were enhanced compared to the untreated group [25].

Therefore, in the present work, we performed a nitrogen (N_2_) plasma surface treatment to improve biocompatibility on the HDPE/n-HAp 3D scaffolds. In addition, we investigated the effect of n-HAp particle adding and nitrogen plasma etching on the 3D HDPE scaffold using preosteoblast (MC3T3-E1) cells.

Preliminary in vitro results will provide the potential of 3D HDPE/n-HAp composite scaffolds in bone reconstruction for clinical applications.

## 2. Materials and Methods

### 2.1. Materials

HDPE and n-Hap (nano powder, <200 nm) were supplied by Sigma-Aldrich (St. Louis, MO, USA). HDPE and n-HAp were used as 3D scaffold materials. 3D HDPE/n-HAp scaffolds were fabricated by 3D printer (3D Bio Printer, M4T-100, M4T Co. Ltd., Daegu, Korea). HDPE polymer mixture was made by mixing the n-HAp concentration of 10 wt%. The HDPE composite was stirred in a heating mantle at 150 °C for 30 min, and then the mixture solution was maintained at room temperature to become a solid state. The composite pieces were filled into the syringe of a 3D printer. HDPE/n-HAp composite was extruded at 170 °C through a nozzle compressed dry air of 580 kPa pressure, and the feed rate was set to 200 mm/min. The nozzle diameter was set to 500 μm and the scaffold struts were deposited layer by layer at angles of either 0° or 90°. 3D scaffold samples with a diameter of 8 mm and 2 mm thickness were used in this study. The pore size of the 3D scaffold was approximately 300 μm.

### 2.2. Nitrogen Plasma Treatment of the 3D HDPE/n-HAp Scaffold

The 3D HDPE/n-HAp scaffold was treated with reactive ion etching (RIE) nitrogen plasma etching due to an increase in the surface roughness and wettability. The N_2_ plasma treatment was carried out by a low-pressure radio frequency (RF, 13.56 MHz) capacitively coupled plasma system (Miniplasma Station, Daejeon, Korea). The N_2_ plasma treatment conditions were performed as follows. The plasma power was set to 100 W, the N_2_ gas flow rates were 20 sccm, and the chamber pressure was set to 1 × 10^−1^ Torr. To achieve the uniform N_2_ plasma treatment to the inside of the 3D HDPE/n-HAp scaffold, the upper layer of scaffold was N_2_ plasma-treated for 5 min the, and then it was turned over and treated for 5 min.

### 2.3. Surface Characterizations

The changes of surface morphology on the untreated and N_2_ plasma-treated 3D scaffolds were observed by field-emission scanning electron microscopy (FE-SEM, S-4800, Hitachi, Tokyo, Japan). Before FE-SEM observation, all samples were deposited by gold sputtering. The FE-SEM observation was performed at an accelerating voltage of 10 kV and micrograph magnifications were to 50×, 1000×, or 5000×.

After the N_2_ plasma treatment, surface topologies of HDPE/n-HAp scaffolds were observed by atomic force microscopy (AFM, XE-100, Park systems, Suwon, Korea) and the images were acquired using XEI software, Version 1.7.6. AFM measurement was carried out at a scanning rate of 0.1 Hz in non-contact mode. Scan areas of 5 μm × 5 μm and 10 μm × 10 μm were randomly selected from the scaffold surface. An arithmetic mean of the root mean square (RMS) roughness (Rq) was calculated directly from the AFM images.

The wettability changes of the N_2_ plasma-treated HDPE/n-HAp scaffold surface were evaluated by static water contact angle (WCA) using a water goniometer (GS, Surface Tech Co., Ltd., Gwangju, Gyeonggi-do, Korea). The contact angle was measured by analyzing the degree of water spreading with charge-coupled device (CCD) digital images after dropping a distilled water droplet (7 μL) on the sample surface. The WCA measurement was repeated five times so as to ensure reliability.

The change of surface chemistry before and after N_2_ plasma treatment were analyzed using Fourier-transform infrared spectroscopy (FT-IR, Spectrum Two FT-IR Spectrometer, Parkin Elmer, Buckinghamshire, UK) in ATR mode, (UATR Two, 4 scan, 4 cm^−1^ resolution). HDPE and HDPE/n-HAp films were used as sample in the FT-IR analysis.

In order to investigate the phase composition and crystallinity of the HAp in the HDPE/n-HAp scaffolds, X-ray diffractometry (XRD, X’Pert PRO MultiPurpose, Philips, Amsterdam, the Netherlands) was used. XRD analysis was performed at 40 kV and 20 mA using Cu-Kα rays. The 2θ scan range was 30–35° and the step size was 0.2°.

All 3D scaffolds were analyzed by X-ray photoelectron spectroscopy (HP-XPS, K-ALPHA+, Thermo Fisher Scientific, Waltham, MA, USA) using a monochromatic Al-Kα X-ray source to obtain their elemental composition and chemical states changed by N_2_ plasma treatment. Each elemental composition of the 3D scaffolds’ surface was reported in atomic percent (at.%) values.

### 2.4. Cell Culture

MC3T3-E1 (newborn-mouse-derived calvaria, CRL-2593) cells were purchased by American Type Culture Collection (ATCC) and cultured in α-Minimum Essential Medium (α-MEM, Gibco, Grand Island, NY, USA) supplemented with 10% fetal bovine serum (FBS, Gibco, Grand Island, NY, USA) and 1% penicillin-streptomycin (Gibco, Grand Island, NY, USA). Cells are cultured in an incubator at 37 °C in a saturated humid atmosphere consisting of 95% air and 5% CO_2_. In vitro evaluation, three passages of MC3T3-E1 cells were used. Before in vitro experiment, the 3D scaffolds were sterilized by ultraviolet C light for 1 h.

### 2.5. Evaluations of Cell Bioactivity

In order to evaluate the proliferation of the MC3T3-E1 cells, 3-(4,5-Dimethylthiazol-2-yl)-2,5-diphenyltetrazolium bromide (MTT, Sigma, St. Louis, MO, USA) assay was used. The 2 × 10^5^ cells/mL were seeded to 3D scaffolds and cultured for 1, 4, and 6 days. The MTT assay is well explained in a previous study [26].

The cell viability was investigated using a live and dead cell staining kit (Biovision, Milpitas, CA, USA). After two days of cell culture on the different sample surfaces, the MC3T3-E1 cells were seeded at a density of 3 × 10^5^ cells/mL on the scaffolds in 48-well plates. After 2 days, the cell culture medium was removed from the 3D scaffolds in 48-well plates and the 3D scaffolds were gently washed with Phosphate-buffered saline (PBS) 3 times. Then, 0.25 mL of the staining solution (1 mM cell-permeable green fluorescent dye and 2.5 mg/mL of propidium iodide) per well and then the 48-well plate was placed in an incubator for 20 min. Live and dead cells images were observed by fluorescence microscope (NI-SS, Nikon, Tokyo, Japan).

The cell differentiation was investigated by alkaline phosphatase (ALP) activity, which is a bone biomarker for bone remodeling processes. Cells were seeded at a density of 1 × 10^5^ cells/mL on 3D scaffolds contained in 48-well plates. After the cells were cultured for 7 and 14 days in osteogenic differentiation media, ALP was measured by quantifying the release of p-nitrophenol (p-NP) from p-nitrophenyl phosphate (p-NPP). The ALP activity assay was well described previous study [26].

In order to investigate the initial adhesion of MC3T3-E1 cells, the cell morphology was observed after culturing of 30 min on the 3D scaffolds. The MC3T3-E1 cells were prefixed with a solution containing 2.5% glutaraldehyde (Sigma-Aldrich) and 2.5% paraformaldehyde (Electron Microscopy Science, Hatfield, PA, USA) for 3 h. The cell fixing method is well described in a previous work [26]. Before SEM observation, fixed cells were coated with a thin gold layer using an automated sputter for 50 s.

## 3. Results

### 3.1. Surface Characterization

Figure 1 shows the FE-SEM micrographs and Energy Dispersive Spectrometer (EDS) spectra on the surfaces of 3D HDPE and 3D HDPE/n-HAp scaffolds before and after nitrogen plasma treatment. In Figure 1a,d,g, the 3D grid structure having a pore size of 300 μm and 0°/90° strut layout pattern is well deposited by 3D HDPE and 3D HDPE/n-HAp using 3D printing. The 3D HDPE and 3D HDPE/n-HAp scaffold showed a relatively flat surface, on the other hand, the 3D HDPE/n-HAp scaffold surface with nitrogen plasma treatment became rougher (Figure 1b,e,h). The EDS spectra demonstrated peaks of calcium and phosphorous in HDPE/n-HAp and N_2_ plasma etched HDPE/n-HAp scaffold compared to HDPE scaffold (Figure 1c,f,i). This result indicated that n-HAp was well incorporated into the HDPE/n-HAp scaffold.

Figure 2 shows the representative AFM 3D-topographical images and their Rq in scan size 15 μm × 15 μm on the 3D HDPE, 3D HDPE/n-HAp, and N_2_ plasma-treated 3D HDPE/n-HAp scaffold. The 3D HDPE and 3D HDPE/n-HAp demonstrated relatively smooth surfaces (Rq = 36.2 nm, 38.1 nm), respectively, as shown in Figure 2a,b. However, after nitrogen plasma etching for 3 min on the 3D HDPE/n-HAp scaffold, scaffold surface was changed to rough surfaces (Rq = 68.1 nm) with repetitive peak-valley structures, as shown in Figure 2c. These results indicate that the surface of the 3D HDPE/n-HAp scaffold was significantly roughened by N_2_ plasma etching. It is thought that n-HAp particle appeared when the HDPE polymer on the surface was etched by nitrogen plasma.

The contact angles of the 3D HDPE, 3D HDPE/n-HAp, and N_2_ plasma-treated 3D HDPE/n-HAp scaffold were shown in Figure 2d. The contact angles of 3D HDPE and HDPE/n-HAp scaffold were 90.96° ± 6.06° and 88.36° ± 3.87°, respectively, showing hydrophobic surface. On the other hand, the 3D HDPE/n-HAp scaffold surface after N_2_ plasma treatment changed to a hydrophilic surface and contact angle was 11.18° ± 2.32°.

FTIR was used to determine the chemical structures of the 3D scaffolds, before and after N_2_ plasma treatment. The FTIR spectra of HDPE, HDPE/n-HAp, and N_2_ plasma treated HDPE/n-HAp are shown in Figure 3. The C-H group of typical bands of HDPE appeared at 2923 and 2853 cm^−1^ were ascribed to asymmetric and symmetric stretching vibrations, respectively [27,28]. In addition, the spectral band at 1468 cm^−1^ was attributed to CH_2_ bending [29]. The very strong peak of hydroxyapatite appeared at 1025 cm^−1^ corresponding to the PO_4_^3−^ functional group [30,31,32]. The peaks of hydroxyapatite were observed in spectra of 3D HDPE/n-HAp, and N_2_ plasma-treated 3D HDPE/n-HAp scaffold. This result means that the n-HAp particle was well incorporated into HDPE polymer. In particular, the appearance of a strong hydroxyapatite peak in the 3D HDPE/n-HAp spectrum is considered to be the etching effect of N_2_ plasma.

Figure 4 shows the XRD patterns of the 3D HDPE, 3D HDPE/n-HAp, and N_2_ plasma-treated 3D HDPE/n-HAp scaffold. Typical HAp peaks were not observed in 3D HDPE scaffold (Figure 4a). However, typical peaks of n-HAp at the 3D HDPE/n-HAp and N_2_ plasma-treated 3D HDPE/n-HAp scaffold clearly appeared as shown in Figure 4b. Furthermore, it showed a similar XRD pattern in the N_2_ plasma-treated 3D HDPE/n-HAp scaffold. The presences of HAp peaks indicate that HAp nanoparticles were well incorporated into the 3D HDPE scaffold.

Element compositions of 3D scaffold surfaces acquired from the wide scan XPS spectra are presented in Figure 5. In the 3D HDPE/n-HAp scaffold, peaks of Ca and P were not observed due to the HAp nanoparticles being covered with an HDPE polymer layer. However, after N_2_ plasma treatment, oxygen peak was increased and peaks of Ca, P, and N were appeared because of the HDPE layer removed by N_2_ plasma etching. It was confirmed again through O1s and N1s analysis of narrow scan XPS spectra.

### 3.2. Evaluations of MC3T3-E1 Cell Bioactivity

The effects of nitrogen plasma treatment and HAp addition on the 3D HDPE/HAp scaffold were evaluated by examining the bioactivity of MC3T3-E1 cells. Physiochemical properties of scaffold surface play an important role in osteoblast cell adhesion, proliferation, differentiation, and bone mineralization.

As presented in Figure 6, the MC3T3-E1 cells cultured for 30 min on the HDPE scaffold were observed round shape and also showed a non-spreading phenotype. However, the cells cultured on the N_2_ plasma-treated HDPE/HAp scaffolds showed the appearance of filopodia spreading and lamellipodia extension and a larger circular shape compared with those on the 3D HDPE scaffolds. For the HDPE/HAp scaffolds, few filopodia spreading and lamellipodia extension were observed in the attached cell morphologies. This result can be explained by the improvement of wettability on the 3D HDPE/n-HAp scaffold by nitrogen plasma treatment enhanced the initial attachment ability compared to 3D HDPE and 3D HDPE/n-HAp scaffolds.

The MC3T3-E1 cell proliferation on different 3D scaffolds was investigated using an MTT assay after being cultured for 1, 4, and 6 days, as shown in Figure 7a. After one day, all samples show no difference in cell proliferation. However, N_2_ plasma-treated HDPE/n-HAp scaffold showed higher MC3T3-E1 proliferation than HDPE and HDPE/n-HAp scaffold at 4 and 6 days. In all the experimental groups, results of the MTT assay demonstrated statistically significant differences compared to the control group (*n* = 3, * *p* < 0.05, ** *p* < 0.01 compared to values shown by 3D HDPE scaffold).

Figure 7b–d shows the live and dead cell staining images of MC3T3-E1 cells cultured for 2 days on 3D scaffolds. The N_2_ plasma-treated HDPE/n-HAp scaffold was observed to have more cells than the 3D HDPE and pristine 3D HDPE/n-HAp scaffolds.

The MC3T3-E1 cell differentiation was investigated using an alkaline phosphatase (ALP) activity (Figure 8). In fact, the ALP is considered a biomarker of osteogenic differentiation and new bone formation. On 7 days, HDPE/n-HAp and N_2_ plasma-treated HDPE/n-HAp scaffold showed higher ALP activity than that of HDPE scaffold. In case of 14 days showed a similar trend. Among the samples, the N_2_ plasma-treated HDPE/n-HAp scaffold had the most differentiation ability compared to other samples.

## 4. Discussion

In recent years, patient-specific bone graft materials to replace defected bone have been extensively fabricated through 3D printing.

The HDPE synthetic polymer is widely used as an implant material for surgical operation due to its hard and non-biodegradable property, which enables it to maintain its position after implantation on the defected region in the oral and maxillofacial trauma. For example, Medpor^®^ composed porous PE has been used in a variety of medical fields such as restoration of cranial defects, facial aesthetic reconstruction, orbital fracture reconstruction, etc. Despite these advantages, few studies on the HDPE 3D scaffold or HDPE/bioceramic composite scaffold deposited by FDM printing have been reported due to the glass transition temperature of HDPE being very high, it is very difficult to control the 3D printing process by melt extrusion. Moreover, the HDPE surface has a hydrophobic nature and it interferes with cell adhesion, resulting in poor integration with bone tissues.

In the present study, we fabricated an HDPE scaffold using FDM 3D printing, and then performed incorporation of n-HAp and N_2_ plasma treatment to enhance the bone regeneration. As shown in Figure 1, we successfully fabricated a 3D HDPE scaffold having a 300 μm and 0/90° strut layout pattern and confirmed the well incorporated n-HAp particle in HDPE polymer through EDS analysis. To overcome the drawback of polymer scaffold itself, research has been extensively conducted on composite scaffolds in which bioceramics are mixed with polymer scaffolds. Among the bioceramic, hydroxyapatite is an essential element for bone tissue regeneration and is well known as a material that promotes osteoblast cell adhesion, osteoconduction, and osteoinduction [33,34,35]. Furthermore, plasma surface modifications have been studied as a means to change the physicochemical surface characteristics including topography, surface charge, and hydrophilicity on the biomaterial surface. Furthermore, it is a useful technique for surface functionalization and surface treatment that introduces functional groups to the surface of materials [36]. For example, nitrogen and oxygen gas plasmas generate polar groups, such as amide, imide, imine, nitrile, hydroxyl, carbonyl, carboxylate, and carboxylic acid groups on the surface of the polymer [37].

After the N_2_ plasma reactive ion etching (RIE), as shown in Figure 2, the contact angle of HDPE/n-HAp scaffolds was significantly decreased and the surface roughness of HDPE/n-HAp scaffolds was increased. This may be thought to be due to the HDPE polymer layer covering HAp nanoparticles incorporated into HDPE/n-HAp scaffolds being removed by N_2_ plasma RIE. Plasma-assisted surface functionalization and plasma etching techniques can provide suitable hydrophilicity and topology changes to the polymeric surface [38,39]. It has been reported that the bone-like apatite layer can provide a suitable surface for osteoblastic growth, proliferation, and differentiation, and this layer enhanced new bone formation [40,41].

Initial cell adhesion after culturing for 30 min on different scaffolds was evaluated by cell morphology using an SEM observation. As observed in Figure 6, N_2_ plasma-treated 3D HDPE/n-HAp scaffold showed well-developed filopodia compared to other 3D scaffolds. This result may be related to the lower contact angle of the N_2_ plasma-treated 3D HDPE/n-HAp scaffold. Many investigators have been investigated the influence of various calcium phosphates on the adhesion, proliferation, and differentiation of bone-related cells [42,43,44,45]. Chou and coworkers reported that HAp enhanced osteoblast differentiation, but more cells adhered to the tissue culture polystyrene surface than to the HAp surface [42]. However, Ogata et al. reported that after culturing for 12 h on each of the types of HAp they tested, there were only ~50 cells/mm^2^ bound to the HAp surface [45]. These contradictory results indicate that while HAp has no cytotoxicity, it does not provide a cell affinity surface. Based on the results of previous studies, it is thought that nitrogen plasma treatment had a significant effect on initial preosteoblast adhesion in this experiment.

In this experiment, proliferation of MC3T3-E1 preosteoblasts were significantly enhanced in the N_2_ plasma-treated 3D HDPE/n-HAp scaffold compared to other scaffold samples. Cell proliferation of 3D HDPE scaffold and 3D HDPE/n-HAp scaffold did not have significant differences, whereas N_2_ plasma-treated 3D HDPE/n-HAp scaffolds showed significant differences (Figure 7). However, cell differentiation of 3D HDPE/n-HAp scaffold was significantly enhanced compared to the 3D HDPE scaffold as shown in Figure 8. Although HAp nanoparticles did not play an important role in cell proliferation, it is thought to have brought about a synergistic effect with nitrogen plasma treatment in cell differentiation.

After N_2_ plasma treatment on the 3D HDPE/n-HAp scaffolds, a new N1s peak appeared at 399.5 eV and O1s peak (531.4 eV) increased as presented in Figure 5. In general, N_2_ plasma generate various nitrogen radicals, such as N_2_, N_2_ (excited), and N [41]. In addition, a polymer surface treated by N_2_ plasma generates nitrogen-containing functional groups such as C—N, C=N, N—O, N=C—O, and N—C=O [46,47,48]. Zhang et al. demonstrated that N_2_ plasma immersion ion implantation (PIII)-treated PE polymer surface has the ability to enhance differentiation of osteoblasts such as ALP activity and Osteocalcin (OC) expressions [49]. Mohammad and co-workers demonstrated that 3D printed PLA scaffold treated RF nitrogen plasma significantly enhanced cell proliferation compared to that of the control sample [50].

Until now, there has been no research on the adhesion, proliferation, and differentiation of preosteoblast on the 3D HDPE/n-HAp scaffold fabricated by FDM 3D printing.

The results of this study are considered to be valuable as a preliminary study for the clinical application of 3D printed HDPE scaffold.

## 5. Conclusions

In the present study, we performed incorporation of n-HAp into 3D HDPE scaffold and then treated N_2_ plasma on 3D HDPE/n-HAp scaffold. Despite the difficulties of the FDM 3D printing process of 3D HDPE scaffold, 3D HDPE/n-HAp composite scaffold with well interconnected pores was successfully fabricated. After N_2_ plasma treatment, the wettability and roughness of the 3D HDPE/n-HAp scaffold surface were enhanced because of the etching effect of N_2_ plasma on the scaffold surface. The HAp addition and N_2_ plasma surface treatment on the 3D HDPE scaffold enhanced the MC3T3-E1 cells’ bioactivities such as initial attachment, proliferation, and differentiation. In particular, it is thought that nitrogen plasma treatment played an important role in showing the highest bioactivity compared to the other scaffolds. From these results, nitrogen plasma treatment may be used as a useful technique for surface modification of HDPE-based bone regeneration scaffolds.

## Figures and Tables

**Figure 1 materials-15-00827-f001:**
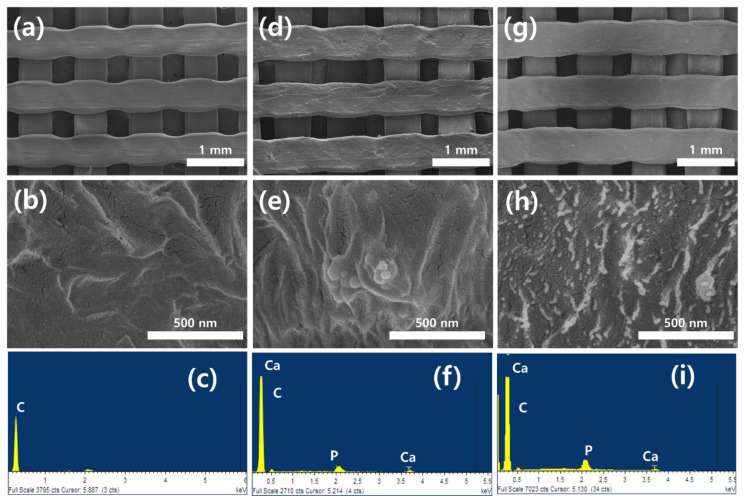
Surface morphologies and Energy Dispersive X-ray Spectroscopy (EDS) spectra on the three dimensional (3D) high-density polyethylene (HDPE) (**a**,**b**,**c**), 3D HDPE/nano-hydroxyapatite (n-HAp) (**d**,**e**,**f**), and Nitrogen (N_2_) plasma-treated 3D HDPE/n-HAp (**g**,**h**,**i**) scaffold.

**Figure 2 materials-15-00827-f002:**
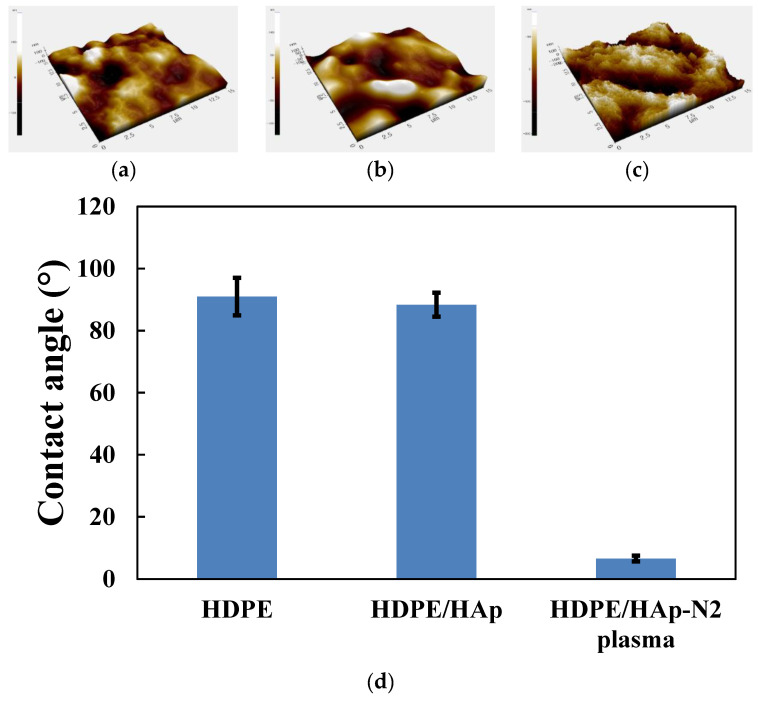
Representative atomic force microscopy (AFM) 3D-topographical images on the (**a**) 3D HDPE, (**b**) 3D HDPE/n-HAp, and (**c**) N_2_ plasma-treated 3D HDPE/n-HAp scaffold. Scan area of all images is 15 × 15 μm. (**d**) Contact angles of 3D HDPE, 3D HDPE/n-HAp, and N_2_ plasma-treated 3D HDPE/n-HAp scaffold surface.

**Figure 3 materials-15-00827-f003:**
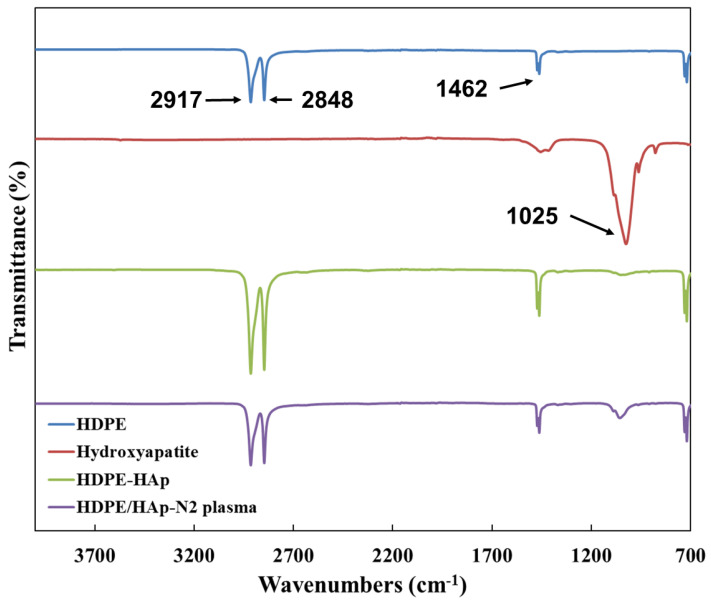
FTIR spectra on the 3D HDPE, 3D HDPE/n-HAp, and N_2_ plasma-treated 3D HDPE/n-HAp scaffold.

**Figure 4 materials-15-00827-f004:**
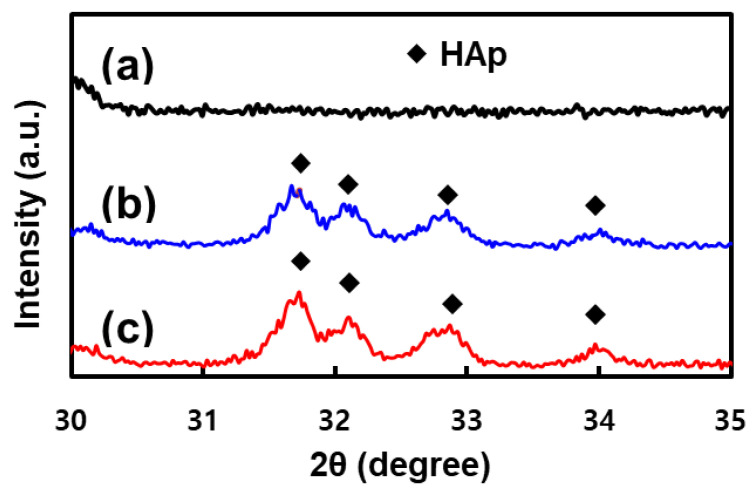
X-ray diffraction (XRD) patterns on the (**a**) 3D HDPE, (**b**) 3D HDPE/n-HAp, and (**c**) N_2_ plasma-treated 3D HDPE/n-HAp scaffold.

**Figure 5 materials-15-00827-f005:**
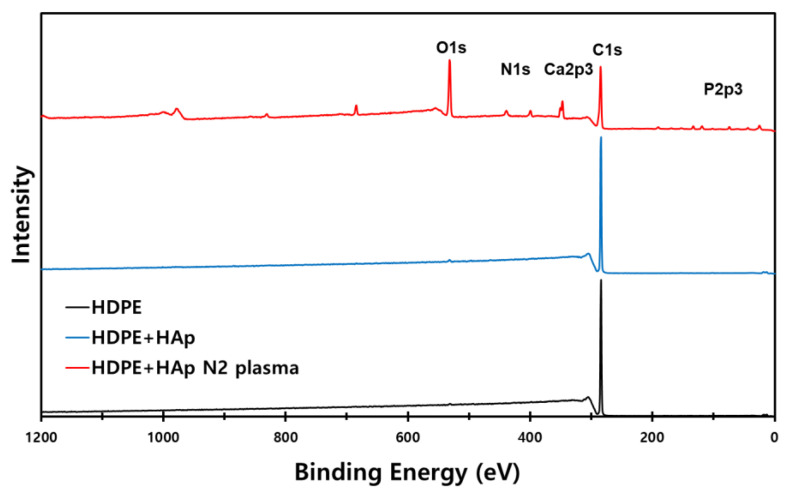
X-ray photoelectron spectroscopy (XPS) wide scan spectra of 3D HDPE, 3D HDPE/n-HAp, and N_2_ plasma-treated 3D HDPE/n-HAp scaffold.

**Figure 6 materials-15-00827-f006:**
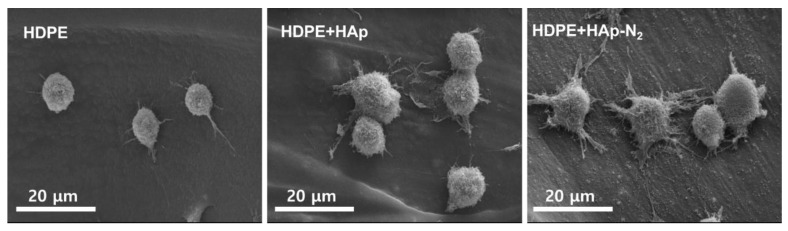
MC3T3-E1 cell morphologies cultured on the 3D HDPE, 3D HDPE/n-HAp, and N_2_ plasma-treated 3D HDPE/n-HAp scaffold for 30 min.

**Figure 7 materials-15-00827-f007:**
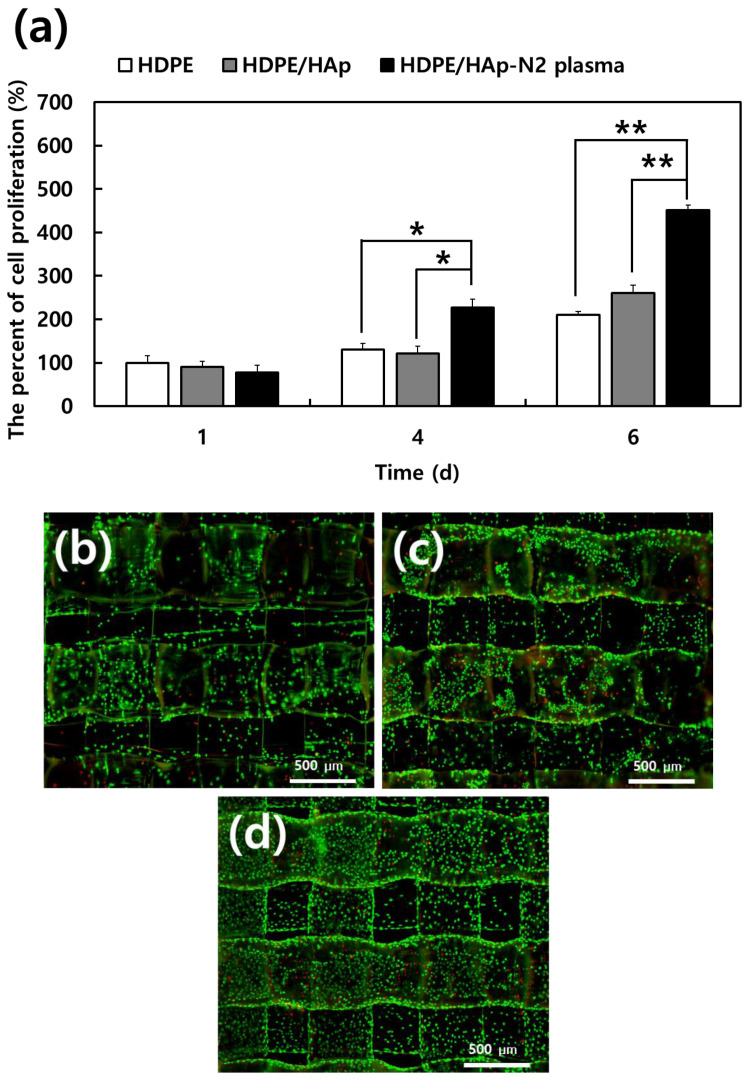
(**a**) Cell proliferation growing cultured on the 3D HDPE, 3D HDPE/n-HAp, and N_2_ plasma-treated 3D HDPE/n-HAp scaffold for 1, 4, and 6 days, as determined by 3-(4,5-Dimethylthiazol-2-yl)-2,5-diphenyltetrazolium bromide (MTT) assay (*n* = 3, * *p* < 0.05, ** *p* < 0.01 compared to values shown by 3D HDPE scaffold; Live and dead cell staining images of MC3T3-E1 cells cultured on (**b**) 3D HDPE, (**c**) 3D HDPE/n-HAp, and (**d**) N_2_ plasma-treated 3D HDPE/n-HAp scaffold cultured for 2 days. Live cells are green and dead cells are red.

**Figure 8 materials-15-00827-f008:**
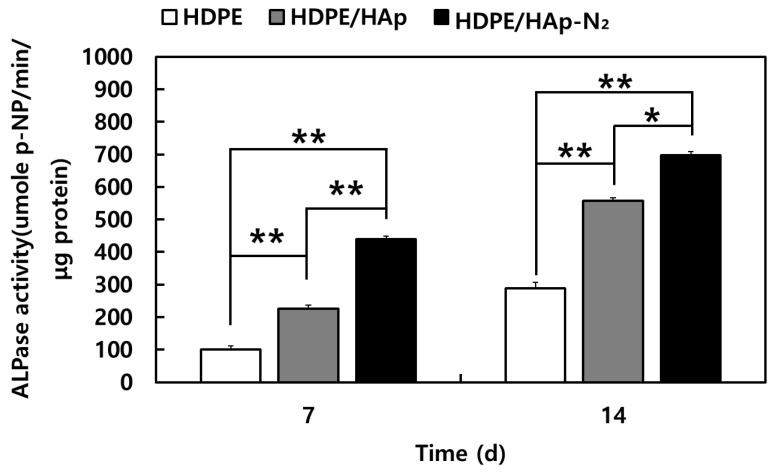
Alkaline phosphatase (ALP) activity of MC3T3-E1 cells cultured for 7 and 14 days on the 3D HDPE, 3D HDPE/n-HAp, and N2 plasma-treated 3D HDPE/n-HAp scaffold for 7 and 14 days. (n = 3, * *p* < 0.05, ** *p* < 0.01 compared to values shown by 3D HDPE scaffold).

## Data Availability

Data sharing not applicable.

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
