# Peer review of "Effect of Hydroxyapatite Nanoparticles and Nitrogen Plasma Treatment on Osteoblast Biological Behaviors of 3D-Printed HDPE Scaffold for Bone Tissue Regeneration Applications"

_materials, 2022, doi:10.3390/ma15030827_

Round 1

Reviewer 1 Report

The manuscript presented by Park et al. discusses the results concerning the use of hydroxyapatite and nitrogen plasma in the functionalization of 3D printed HDPE scaffolds for biomedical applications. The Authors used several suitable techniques to study how the presence of n-HAp and plasma treatment influences the cell bioactivity. First of all, the extensive editing of English language and style is required. Some minor and major mistakes are present all over the manuscript (for example: to improving, was appeared, etc.). My comments to the manuscript are presented below. I recommend reconsidering the presented manuscript to be published after major revision.

Minor comments:

  • The presented paper is based on quite old references – most of the cited works are older than 10 years. Therefore, the Authors should consider more recent works relevant to the field of study. Moreover, the Introduction is rather short and not much novelty is shown here.
  • The EDS spectra images are of low quality. The elements are barely visible. This must be improved.
  • The Authors analysed “The crystallography” of the samples. According to Wikipedia: “Crystallography is the experimental science of determining the arrangement of atoms in crystalline solids”. Therefore, nor the crystallography, but rather XRD was used to analyse the phase composition and crystallinity of the samples. Please rephrase.

Major comments:

  • As the Authors state on page 10 “In general, N2 plasma generate the various nitrogen radicals, such as N2, N2 (excited) and N [41]. In addition, N2 plasma is possible to produce nitrogen-containing functional groups, such as C–N, C=N, N–O, N=C–O and N–C=O, on a polymer surface [39–41].”, shouldn’t the additional nitrogen be visible on the EDS spectra of the studied samples after the N2-plasma activation?
  • The Authors discussed the wettability of the prepared samples using the contact angles measurements. However, the Methods section is missing information about that kind of measurements. Moreover, no information whether the Authors studied the advancing or receding contact angles values were shown. Furthermore, I also couldn’t find the information what kind of solvent was used in the study. Since this is important to draw conclusions regarding the surface hydrophobicity, this kind of information must be included in the manuscript.

Reviewer 2 Report

In the present paper, the fabrication of 3-dimensional scaffolds based on high-density polyethylene (HDPE) and nanohydroxyapatite (n-HAp) is described. The surface of the scaffolds is further functionalized/modified via nitrogen plasma, which clearly improves cell adhesion, proliferation and differentiation. Even if the polymer matrix (i.e., HDPE), the filler (i.e., n-HAp) or the manufacturing process (i.e., fused deposition modelling) do not represent a clear advance in the current state of the art, the present manuscript presents useful information and a significant improvement with respect to clinically available alternatives (i.e., Medpor®). Several aspects should be considered to increase the robustness of the present manuscript, as detailed below:

- Introduction: References highlighting other polymer/bioceramic composites for bone tissue engineering are missing to support the following statement: “Recently, synthetic polymer/bioceramics scaffolds were widely used as alternative bone grafting materials in bone tissue engineering, in which bioactive ceramic particles are incorporated.” See, for example: i) https://doi.org/10.1016/S0142-9612(02)00131-X, ii) https://doi.org/10.1002/jbm.a.35525, iii) https://doi.org/10.1002/term.249.

- In section “2.5. Evaluations of cell bioactivity”, cell densities should be normalized. Sometimes, authors refer to cell/well, other times to cell/ml. Referring to cells/scaffold may be the most appropriate alternative in this case.

- In section “2.5. Evaluations of cell bioactivity”, the paragraph about cell viability is confusing. Particularly, the following sentence should be rewritten: “After two days of cell culture on the different sample surfaces, the MC3T3-E1 cells were seeded at a density of 3 × 105 cells/mL on the scaffolds in 48-well plates.”

- To gain more information about the surface chemistry after N2 plasma treatment, X-ray photoelectron spectroscopy (XPS) should be employed. In fact, in Figure 1, the EDS spectrum was unable to confirm the presence of nitrogen functionalities on the surface of the plasma-treated scaffold.

- The presence of hydroxyapatite in the scaffolds is well demonstrated by XRD, FTIR and EDS. However, a thermogravimetric analysis could provide more information about the quantity (weight%) that is really incorporated.

- Regarding the MTT assay, in the methodology it is stated that the test was performed after 1, 3 and 5 days. However, in the results section, the time-points were 1, 4 and 6 days. In this same section, authors claim that “However, N2 plasma-treated HDPE/n-HAp scaffold showed higher MC3T3-E1 proliferation than pristine PCL and HDPE/n-HAp scaffold at 4 and 6 days.” PCL should be corrected and substituted by HDPE.

- The manuscript should be revised by a native speaker to correct several grammatical and typographical errors.

Round 2

Reviewer 1 Report

Dear Authors,

thank you very much for your thorough revision of your manuscript. I have no further comments for this matter, therefore I recommend to accept the paper in the present form.